# Use of Antibiotics in Poisonous Ingestions of Corrosives and Organophosphates: A Retrospective Cohort Study

**DOI:** 10.3390/toxics11040300

**Published:** 2023-03-24

**Authors:** Joud K. Altuwaijri, Fatma M. Hamiduddin, Raghad H. Khafaji, Leyan T. Almaghrabi, Hussain T. Bakhsh, Abrar K. Thabit

**Affiliations:** 1Faculty of Pharmacy, King Abdulaziz University, Jeddah 22254-2265, Saudi Arabia; 2Pharmacy Practice Department, Faculty of Pharmacy, King Abdulaziz University, Jeddah 22254-2265, Saudi Arabia

**Keywords:** antibiotics, corrosive, organophosphates, poisoning, toxicity, ingestion, infection

## Abstract

The use of antibiotics following oral poisoning by corrosives and organophosphates is controversial. We assessed the clinical outcomes of using antibiotics in acute poisonous ingestion involving corrosives or organophosphates by conducting a retrospective cohort study of patients presenting to the emergency department following ingestion of corrosives or organophosphates who received either antibiotics or supportive care. The endpoints included clinical stability, length of stay (LOS), and mortality. Of 95 patients, 40 received antibiotics and 55 received supportive care. The median age was 2.1 and 2.7 years, respectively (*p* = 0.053). Bacterial growth was shown in only 2 of 28 cultures (both were respiratory), but with hospital-acquired organisms as it was shown ≥4 days post-admission. Clinical stability rates were 60% and 89.1% in the antibiotic and supportive care groups, respectively (*p* < 0.001). Median LOS was 3 vs. 0 days (*p* < 0.001), and no mortality was recorded. NG/G-tube placement was the only factor associated with clinical failure (OR, 20.97; 95% CI, 2.36–186.13). Antibiotic use was not associated with higher chances of clinical stability, which may suggest that their use was unnecessary. Clinicians are encouraged to use antibiotics wisely, and only in the presence of a clear indication of an infection. This study provides a basis for future prospective studies to confirm its findings.

## 1. Introduction

Human poisoning by toxic ingestion is very common; however, the lack of standard guidelines or a common consensus for the management of such incidents while considering different patients’ characteristics may lead healthcare providers to rely on the available literature, as well as their judgment and expertise, in deciding the best course of treatment to manage the patients’ symptoms and prevent complications [1]. While poisonous substances may differ, corrosives and organophosphates in particular are known to cause chronic complications; hence, they trigger clinicians to adapt more aggressive management plans beyond supportive care [2,3,4].

Corrosives are substances that cause damage and corrosion by chemical action, causing tissue necrosis at pH levels that are considered highly corrosive (<2 or >12). Approximately 30% of corrosive ingestion cases occur in children, resulting in an estimated global mortality rate of 4.8 per 100,000 individuals per year [5]. Corrosives can also lead to complications, such as gastrointestinal (GI) tract injuries, perforations, and strictures [4]. Treatment methods for poisoning due to corrosive ingestion remain debatable; however, antibiotics have been used despite the insufficient number of controlled studies to confirm their role in such cases [4].

Organophosphates are chemicals produced by the esterification of phosphoric acid and alcohol [6]. They are the main component of herbicides, pesticides, and insecticides. Global data from the literature and the World Health Organization cause-of-death database indicate that about 385 million cases of unintentional poisoning by pesticides occur annually [7]. If absorbed systemically, organophosphates have the ability to irreversibly bind to acetylcholine esterase, preventing the breakdown of acetylcholine and resulting in the overstimulation of muscarinic and nicotinic receptors (Figure 1) [6]. This overstimulation causes organophosphate complications that affect different body systems, such as the respiratory system (aspiration pneumonia from excessive salivation, respiratory failure due to muscle weakness, and bronchospasms), the cardiovascular system (arrhythmia, prolonged QTc interval, bradycardia, and blood pressure dysregulation) and the central nervous system (seizures and mental status changes) [6]. Definitive therapy for organophosphate toxicity can be achieved using atropine and pralidoxime to reverse their effects at the muscarinic and nicotinic receptors, respectively [8].

The legitimacy of using antibiotics in cases of toxic ingestion can stem from concern regarding bacterial complications, such as aspiration pneumonia with organophosphates or perforation after corrosive ingestion. Recent intra-abdominal infection guidelines stated that both gastroduodenal and small bowel perforations are a common cause of peritonitis; thus, antibiotic therapy would be indicated. Recommended antibiotics included amoxicillin/clavulanate, piperacillin/tazobactam, and ceftriaxone. Alternatively, fluoroquinolones or aminoglycosides can be used for patients with severe β-lactam allergies [9], although fluoroquinolones are generally avoided in pediatrics due their adverse effect on skeletal development [10]. The recommended duration is four days in immunocompetent, stable patients, and longer in the presence of signs of persistent infection [9].

Due to the lack of studies confirming the role of antibiotic therapy in toxic ingestion poisoning, this study aimed to assess the clinical outcomes of using antibiotics in cases of acute poisonous ingestion involving either corrosives or organophosphates. The results of this study may provide a basis on which future prospective studies may be developed to confirm its findings.

## 2. Results

### 2.1. Baseline Characteristics

A total of 95 patients were eligible for study inclusion; 40 were included in the antibiotics group and 55 in the supportive care group. No difference was observed between the groups in median age, gender, distribution of the type of toxic ingestion, presence of leukocytosis, or development of GI bleeding (Table 1). However, differences were observed in the distribution of age groups (most patients were pediatric); number of patients admitted to the intensive care unit (ICU); incidence of fever within 48 h of admission, despite being present in only a small subset of patients; number of patients from whom a microbiological culture was collected; number of patients who underwent endoscopy; and number of patients who had a nasogastric (NG) tube or gastronomy tube (G-tube) placed. The majority of patients in both groups reported poisoning by a corrosive substance (90% vs. 74.5%) rather than by an organophosphate (10% vs. 25.5%), but the difference in the distribution of patients was not significant (*p* = 0.058). The most commonly prescribed antibiotic in the antibiotics group was amoxicillin/clavulanic acid, which was prescribed to 32 of 40 of the patients (80%). The fluoroquinolone moxifloxacin was prescribed to only one patient, who was eleven years old, and it was used for only three days before it was discontinued by the infectious diseases consultant and switched to meropenem.

### 2.2. Microbiological Findings

While microbiological cultures were collected from a total of 17 patients (13 vs. 4), some patients had more than 1 culture collected from them. The total number of cultures collected from these patients was 28, all from different sites (Table 1). Although only two cultures showed growth, both of these were of respiratory origin. One culture showed *Pseudomonas aeruginosa,* while the other showed the same organism with carbapenem-resistant *Klebsiella pneumoniae*. Notably, these two cultures were collected at 4 days and 8 days post-admission from a patient who had ingested a corrosive (a household detergent) and a patient who was intoxicated with an organophosphate, respectively.

### 2.3. Endoscopic Findings

The patients who underwent endoscopy (n = 52) were all from the group of patients who had ingested a corrosive substance. Among the 31 patients in the antibiotics group, 10 (32.3%) had normal findings, while the remaining 21 patients had findings ranging from mild esophageal, gastric, and/or duodenal ulceration to esophageal stricture or fistula. Moreover, some patients experienced tonsil or oropharyngeal ulceration. On the other hand, of the 21 patients in the supportive care group, 9 (42.9%) had normal findings, whereas 12 patients experienced pharyngeal, esophageal, gastric, and/or duodenal ulceration. One patient developed an esophageal obstruction, while another developed a vocal cord edema.

### 2.4. Clinical Outcomes

Table 2 lists the clinical outcomes of the patients. The clinical stability rate was significantly lower in the antibiotics group than in the supportive care group (60% vs. 89.1%; difference of 29.1%; *p* < 0.001). However, clinical failure was attributed to toxicity-related complications rather than infections, such as esophageal strictures or cardiac complications. Only three patients, however, developed chemical pneumonitis post-presentation that required antibiotic therapy. Although no mortality was recorded in either group, patients in the antibiotics group tended to stay longer in the hospital, with a median (IQR) LOS of 3 (1–7.8) days vs. 0 (0–2) days in the comparator group (*p* < 0.001).

### 2.5. Factors Associated with Clinical Failure

The factors that were investigated to be associated with clinical failure using multivariate regression included receipt of antibiotics, ICU admission, presence of fever, GI bleeding, and placement of NG or a G-tube. Of these factors, only the placement of NG or a G-tube was significantly associated with clinical failure (OR, 20.97; 95% CI, 2.36–186.13; *p* = 0.006).

## 3. Discussion

Poisoning by ingestion is a prevalent problem; however, healthcare providers tend to depend primarily on the available literature and their own clinical judgment to determine the best course of management. Corrosives and organophosphates are known to induce chronic complications that might result in unfavorable outcomes and, thus, should be addressed carefully. In this study, we aimed to examine the clinical outcomes when antibiotics were used in cases of acute toxic ingestion involving corrosives or organophosphates.

Our study demonstrated that patients who received antibiotics had significantly lower clinical stability rates and longer lengths of stay than patients who did not. Such findings were is in line with the findings of Priyendu et al. regarding patients with organophosphate poisoning; they reported that most of those who received antibiotics had unfavorable outcomes and experienced more negative effects compared to those who did not receive them [3]. Despite the absence of conclusive evidence to support the use of antibiotics as an intervention in cases of poisoning, we found that antibiotics were used in nearly half (42%) of the patients. Based on an untested “surgical myth” dating back more than 60 years, antibiotics can be used during the acute stage of burns, as they may help to prevent stricture development [11]. However, one study found that using steroids and antibiotics not only increased the likelihood of perforation, but also failed to diminish stricture development [12]. In a review of the literature concerning corrosive substance injuries, it was suggested that prophylactic antibiotics could be administered to patients with suspected perforations and grade 3 injuries (defined as multiple ulcerations and areas of brown-black or greyish discoloration, suggesting necrosis) despite stating that such a suggestion is not validated by scientific evidence [13]. Notably, all the patients in the antibiotics group were pediatrics, which may explain why clinicians opted to prescribe antibiotics to this vulnerable population; they feared the development of further complications. Nevertheless, given the findings of this study and the study by Priyendu et al., which was performed on adults, the use of antibiotics could not be justified regardless of the age group of the patients [3].

While microbiological cultures were collected from only 17 of 95 patients in our cohort (17.9%), with a total of 28 cultures from three major sites (blood, respiratory, and urine), most of these cultures were collected from the patients in the antibiotics group, of which 10 (25.6%) patients in the group also had fever vs. only 3 (7.7%) in the supportive care group. This may also explain why antibiotics were administered to those patients. The very small number of patients from whom a culture was collected may indicate that there was probably no clear trigger for microbiological culture collection to confirm the presence of an infection. Moreover, none of the cultures showed bacterial or fungal growth, except for two respiratory cultures which showed growth of bacteria that are known to be hospital-acquired (*P. aeruginosa* and carbapenem-resistant *K. pneumoniae*). These cultures were collected a few days after hospital admission; hence, they were less likely to be of the patients’ normal flora since this was the patients’ first hospital admission. A study by Kalayarasan et al. found that the routine use of broad-spectrum antibiotics was not indicated, except in patients with sustained high-grade injuries and those receiving systemic steroids [14]. Furthermore, given that it may be impossible to discriminate between the effects of antibiotics vs. steroids, it may also be a challenge to justify the use of antibiotics in cases of poisoning [15]. In their review of corrosive substance ingestion, Hall, et al. concluded that antibiotics can have therapeutic effects in events of perforation and infection, but not as prophylactics [5].

In our examined data, ICU admission rates were higher among the patients who received antibiotics (22.5%). This indicates that they were sicker than the patients in the supportive care group; thus, they had higher rates of complications and clinical failure. The administration of antibiotics to the critically ill patients in the antibiotics group in our study was expected, as previous studies have shown that antimicrobial prophylaxis accounts for a large portion of antibiotic prescriptions in the ICU [16]. It should be noted, however, that the unnecessary prescription of antibiotics (i.e., with a lack of evidence of sepsis) can have serious consequences, including increased risk of antimicrobial resistance, excessive side effects, and increased healthcare costs [17].

Our findings indicate that co-amoxiclav (amoxicillin/clavulanic acid) was the most frequently prescribed antibiotic, being given to 80% of the patients in the antibiotics group. Given that the majority of the poisoning was corrosive in nature, this finding may be related to the fact that perforation is a frequent complication of corrosive poisoning and co-amoxiclav is one of the agents that is recommended in the intra-abdominal infection guidelines to be used in such an indication, given its broad-spectrum coverage of Enterobacterales and anaerobes [9]. Our study also revealed that the antibiotic group underwent significantly more endoscopies than the supportive care group. This could be explained by the fact that most patients in the antibiotics group (90%) had corrosive toxicity, for which endoscopy is part of the standard of care to investigate the presence of perforation and administer antibiotics accordingly [18].

The decision to place an NG or G-tube is typically based on the patient’s condition and whether gastric lavage/aspiration, bowel irrigation, or flushing of toxic material was needed [19]; nonetheless, clinicians should be aware that placing a tube can augment the risk of the formation of long strictures, which may further complicate the situation [14]. As a result, routine NG or G-tube placement with the aim of evacuating any remaining corrosive material is no longer justified prior to endoscopic assessment of mucosal injury [15]. This is attributed to the high risk of vomiting or retching, which may lead to additional esophageal exposure through reflux of the residual intragastric corrosive substance. Furthermore, insertion of a foreign object into the acute inflamed environment may promote the development of infection, thus delaying mucosal healing [15]. These consequences were consistent with our finding that the placement of an NG or G-tube was significantly associated with clinical failure.

While the lack of benefit of antibiotics was shown in a previous study concerning organophosphate poisoning, this is the first study that also involved assessment of the use of antibiotics in cases of acute poisoning by corrosive substances. However, a few limitations are recognized. Given the retrospective nature of the study, some data were missing in the electronic medical records due to poor documentation in the progress notes or neglect to order some laboratory tests (complete blood counts or microbiological cultures). Those patients, as well as patients who were deceased within 24 h of presentation, were excluded, as the outcome in question (clinical stability) could not be identified nor evaluated. Lack of data was identified as an exclusion criterion in compliance with previous guidelines for retrospective chart review studies [20,21]. Such an issue could be overcome by a well-designed prospective study. In addition, all data were revised by the corresponding author to ensure that no selection bias occurred. Furthermore, the small sample size, as well as the fact that the study was conducted in a single center, may limit the generalizability of its findings. Therefore, larger studies are warranted to confirm the findings of the current study.

## 4. Conclusions

In conclusion, corrosive and organophosphate poisoning continues to be an important healthcare issue, and neither explicit guidelines nor sufficient clinical data are available to assist healthcare practitioners regarding the treatment approach. This study showed that the use of antibiotics in cases of toxic ingestion of corrosives or organophosphates was not associated with increased chances of clinical stability compared with supportive care alone. According to the findings of this study, we suggest that there is no need for prophylactic antibiotic treatment. As such, clinicians are encouraged to prudently use antibiotics only in the presence of a clear indication of an infectious process or signs of sepsis, such as fever, leukocytosis, elevated inflammatory markers, signs of inflammation on radiological imaging, or bacterial growth in relevant cultures (blood in case of perforation and respiratory in case of pneumonitis). The results from this study can provide a basis for future prospective studies to confirm its findings.

## 5. Materials and Methods

### 5.1. Study Design and Patients

This was a retrospective cohort study intended to evaluate the clinical outcomes of the use of antibiotics compared with supportive care in cases of poisoning caused by corrosives or organophosphates. Supportive care was defined as symptomatic therapy and a standard of care that did not involve the use of antibiotics. We included all patients of any age who presented to the emergency department (ED) from November 2010 to October 2021 if they had ingested corrosives or organophosphate and had received either antibiotics or supportive care. Patients who passed away within 24 h or lacked sufficient data in their electronic medical records were excluded. This study was approved by the Biomedical Research Ethics Committee of the Faculty of Medicine (approval No. 416–421).

### 5.2. Endpoints

The primary endpoint was clinical stability, which was defined as the resolution of initial signs and symptoms of toxicity without the development of infectious complications. This was confirmed by progress notes and discharge notes. Secondary endpoints were the length of hospital stay (LOS) and in-hospital mortality. Moreover, bacterial growth was assessed for patients from whom microbiological cultures were collected (from any site) within seven days of presentation to the ED.

### 5.3. Statistical Analysis

Data were compared using the Chi-square test or the Mann–Whitney U test for categorical and continuous variables, respectively. Categorical variables were presented as numbers (percentages), whereas continuous variables were presented as medians (interquartile range, IQR). The distribution of the variables was confirmed using Shapiro–Wilk’s test of normality. Statistical significance was indicated by a two-sided *p* value of <0.05. Multivariate logistic regression was conducted to investigate the factors associated with clinical failure, and only factors with a *p* < 0.1 in the univariate analysis were included. All analyses were conducted using SPSS version 28.0 (IBM, Corp., Armonk, NY, USA).

To achieve a power of 80%, a total of 94 patients were needed. This number was based on an estimated difference in the primary outcome of 30% between the two groups and an α error probability of 0.05. An effect size of 30% was selected based on a previous study on antibiotic prophylaxis in organophosphate poisoning, which showed a difference in unfavorable outcome rates of approximately 40% between the antibiotics group and the group that did not receive antibiotics [3]. This number was lowered to 30% in order to expand the sample size.

## Figures and Tables

**Figure 1 toxics-11-00300-f001:**
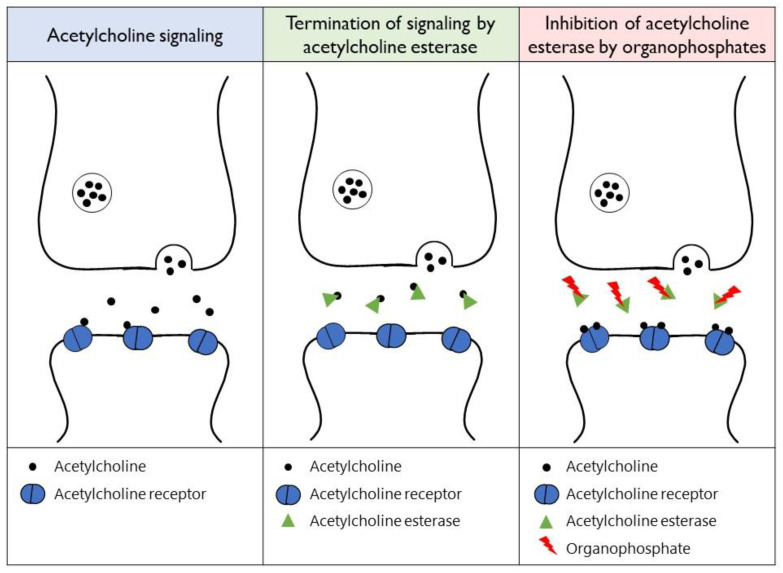
An illustration of organophosphate inhibition of acetylcholine esterase and the resultant overstimulation of acetylcholine receptors at the muscarinic and nicotinic receptors.

**Table 1 toxics-11-00300-t001:** Baseline characteristics of patients.

Characteristic	Antibiotics Group (n = 40)	Supportive Care Group (n = 55)	*p* Value
Age (years)	2.1 [1.2–4.3]	2.7 [2–10]	0.053
Age group			0.012
Pediatric (<18 years)	40 (100)	47 (85.5)	
Adult (≥18 years)	0 (0)	8 (14.5)	
Sex (male)	23 (57.5)	31 (56.4)	0.912
Type of poisoning			0.058
Corrosive	36 (90)	41 (74.5)	
Organophosphate	4 (10)	14 (25.5)	
Type of Corrosive			0.058
Household detergent	25 (69.4)	16 (50)	
Bleach	1 (2.8)	8 (25)	
Soap	1 (2.8)	1 (3.1)	
Unspecified	5 (13.9)	6 (18.8)	
Other	4 (11.1)	1 (3.1)	
Admitted to the ICU	9 (22.5)	3 (5.5)	0.014
Fever ^a^	10 (25.6)	3 (7.7)	0.033
Leukocytosis ^b^	21 (53.8)	17 (44.7)	0.424
Culture collection within 7 days	13 (32.5)	4 (7.3)	0.002
Sites of cultures ^c^			0.706
Blood			
Respiratory			
Urine			
Growth in culture ^d^	2 (9.1)	0 (0)	0.567
Gastrointestinal bleeding	2 (5)	1 (1.8)	0.381
Endoscopy performed	31 (77.5)	21 (38.2)	<0.001
NG or G-tube placement	9 (22.5)	0(0)	<0.001
Antibiotic received ^e^		NA	NA
Co-amoxiclav	32 (80)		
Ceftriaxone	4 (10)		
Meropenem	3 (7.5)		
Cefuroxime	3 (7.5)		
Metronidazole	3 (7.5)		
Vancomycin	2 (5)		
Erythromycin	1 (2.5)		
Moxifloxacin	1 (2.5)		
Days to antibiotic initiation	0 [0–1]	NA	NA
Antibiotic duration (days)	4 [2.25–8]	NA	NA

Data are presented as n (%) or median (interquartile range); G-tube, gastronomy tube; ICU, intensive care unit; NG, nasogastric; ^a^ temperature > 37.5°; ^b^ white blood cells count of >10,500 cells/mm^3^; ^c^ some patients had more than one culture collected from different sites with a total number of cultures of 28. ^d^ Of the total number of cultures collected from patients in each respective group, ^e^ some patients received more than one antibiotic.

**Table 2 toxics-11-00300-t002:** Outcomes of patients.

Characteristic	Antibiotics Group (n = 40)	Supportive Care Group (n = 55)	*p* Value
Clinical stability	24 (60)	49 (89.1)	<0.001
Length of stay	3 [1–7.75]	0 [0–2]	<0.001
In-hospital mortality	0 (0)	0 (0)	NA

Data are presented as n (%) or median (interquartile range).

## Data Availability

Data are available from the corresponding author upon request.

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
