# Peer review of "Use of Antibiotics in Poisonous Ingestions of Corrosives and Organophosphates: A Retrospective Cohort Study"

_toxics, 2023, doi:10.3390/toxics11040300_

Round 1

Reviewer 1 Report (Previous Reviewer 1)

I consider that the clarifications have been correct and I do not find particularly relevant reasons to reject this article.

Author Response

Thank you very much for the positive feedback.

Reviewer 2 Report (New Reviewer)

Use of antibiotics in poisonous ingestions of corrosives and or ganophosphates: A retrospective cohort study.

Authors affirmed: "No mortality was recorded". May be this data could have affect the results? Was this data excluded by Authors?m please explain.

Human poisoning by toxic ingestion is very common; data worlwide?

 lack of standard  guidelines or a common consensus for management may lead healthcare providers to rely  on available literature, as well as their judgment and expertise in deciding the best course  of treatment to manage the patients’ symptoms and prevent complications: this is a crucial point, though, is worthy of saying guidelines are not "protocol" and must be not applied within circumstances and patients' carachteristics (Please consider: Guidelines and current assessment of health care responsibility in Italy.  Zerbo, S., Malta, G., Argo, A.  Risk Management and Healthcare Policy, 2020, 13, pp. 183–189).  

The decision to place NG or G-tube is typically based on the patient's condition; nonetheless, clinicians should be aware that placing a tube can augment the risk of the formation of long strictures, which may further complicate the situation [11]. Which is the causal relationship  within this point in regard to the use of antibiotics in poisonous ? Please clarify. 

The size number of patient of 2 group seems adequate.   

Author Response

  1. Authors affirmed: "No mortality was recorded". May be this data could have affected the results? Was this data excluded by Authors? please explain.

Response: We included all eligible patients according to the inclusion and exclusion criteria listed on lines 265-269. We excluded those who passed away within 24h since the presentation. All data were revised by the principal investigator (the corresponding author) to ensure no selection bias has occurred. To address this comment, we added this statement to the limitations part of the discussion (lines 243-244).

Patients who deceased within 24h and those with insufficient data in their medical records do not have enough information to appropriately evaluate the primary outcome of the study (i.e., clinical stability). We carefully evaluated previously published guidelines for conducting retrospective chart reviews and found that excluding such cases is suggested. Per your suggestion, we added a justification and clarification regarding this issue to the limitations part of the discussion (lines 240-242) with added citations to the guidelines, which are:

https://onlinelibrary.wiley.com/doi/10.1046/j.1524-4733.2003.00242.x (here, please refer to the section titled “Event capture: are the data, as collected, able to identify the intervention and outcomes if they actually occurred?”)

and

https://www.ncbi.nlm.nih.gov/pmc/articles/PMC3853868 (here, please refer to point 7 “Failure to explicitly develop inclusion and exclusion criteria”, where the authors state “Suggestions for exclusion criteria include sufficient lack of variables recorded in the chart, …”).

  1. Human poisoning by toxic ingestion is very common; data worldwide?

Response: Epidemiological data were added to lines 46-48 and 55-57.

  1. Lack of standard guidelines or a common consensus for management may lead healthcare providers to rely  on available literature, as well as their judgment and expertise in deciding the best course  of treatment to manage the patients’ symptoms and prevent complications: this is a crucial point, though, is worthy of saying guidelines are not "protocol" and must be not applied within circumstances and patients' characteristics (Please consider: Guidelines and current assessment of health care responsibility in Italy.  Zerbo, S., Malta, G., Argo, A.  Risk Management and Healthcare Policy, 2020, 13, pp. 183–189).

Response: Thank you for the suggestion. We reviewed the suggested citation and agree that it is appropriate to cite it here (line 41). We also slightly edited the sentence to appropriately address your comment (lines 38-39).

  1. The decision to place NG or G-tube is typically based on the patient's condition; nonetheless, clinicians should be aware that placing a tube can augment the risk of the formation of long strictures, which may further complicate the situation [11]. Which is the causal relationship within this point in regard to the use of antibiotics in poisonous? Please clarify.

Response: NG tube placement is often needed to flush away the gastric content or aspirate the content. A clarification was added to lines 221-222.

  1. The size number of patient of 2 group seems adequate.

Response: Thank you for confirming this.

Sincerely,

The corresponding author

This manuscript is a resubmission of an earlier submission. The following is a list of the peer review reports and author responses from that submission.

Round 1

Reviewer 1 Report

It does not seem that the study is really a retrospective cohort study, it is necessary to justify it better.

Line 47 - The “thermal skin burns” clarification is unnecessary.

Line 67 - Clarify in which patients quinolones were used, since most of the 2 cohorts are made up of pediatric patients where they are avoided due to adverse effects.

Line 80 - Clarify what patients refer to in "supportive care", is it only symptomatic treatment?

Line 232 - Such a categorical statement cannot be made with this study, it would be more appropriate "According to what has been observed, it is suggested that there is no need for prophylactic antibiotic treatment."

Line 247 - It seems to me a great source of bias that deceased patients or those who did not have an electronic medical record were not included. Justify the reason for this decision.

Author Response

Thank you for reviewing our manuscript and for your valuable comments. Please find below the responses to your comments.

  1. It does not seem that the study is really a retrospective cohort study, it is necessary to justify it better.

Response: As this was an outcomes study with retrospective chart review, we believe that it meets the definition of a “retrospective cohort study” as defined by the American National Cancer Institute of the National Institute of Health: https://www.cancer.gov/publications/dictionaries/cancer-terms/def/retrospective-cohort-study

However, to address this comment, we rephrased the first sentence under study design to better reflect this definition (lines 253-255).

  1. Line 47 - The “thermal skin burns” clarification is unnecessary.

Response: Thank you for the note. This was removed as suggested.

  1. Line 67 - Clarify in which patients quinolones were used, since most of the 2 cohorts are made up of pediatric patients where they are avoided due to adverse effects.

Response: The statement that the reviewer is referring to is in the introduction. Therefore, we added a statement regarding the unfavorable use of fluoroquinolones in pediatrics with a cited reference (lines 69-70). In our study, we only had one patient receiving a fluoroquinolone (moxifloxacin) as shown in Table 1 who was 11 years old. We returned to the patient’s medical record to evaluate this use and found that the patient received it for only 3 days before the intervention of the infectious diseases team who discontinued it and started meropenem instead. A statement on this was added to the results section (lines 93-95).

  1. Line 80 - Clarify what patients refer to in "supportive care", is it only symptomatic treatment?

Response: Yes. This was clarified under “Study design and patients” on lines 256-257.

  1. Line 232 - Such a categorical statement cannot be made with this study, it would be more appropriate "According to what has been observed, it is suggested that there is no need for prophylactic antibiotic treatment.”

Response: Thank you for the suggestion. The statement was edited accordingly (lines 243-245).

  1. Line 247 - It seems to me a great source of bias that deceased patients or those who did not have an electronic medical record were not included. Justify the reason for this decision.

Response: We appreciate your comment; however, patients who deceased within 24h and those with insufficient data in their medical records do not have enough information to appropriately evaluate the primary outcome of the study (i.e., clinical stability). We carefully evaluated previously published guidelines for conducting retrospective chart reviews and found that excluding such cases is suggested. Per your suggestion, we added a justification and clarification regarding this issue to the limitations part of the discussion (lines 229-233) with added citations to the guidelines, which are:

https://onlinelibrary.wiley.com/doi/10.1046/j.1524-4733.2003.00242.x (here, please refer to the section titled “Event capture: are the data, as collected, able to identify the intervention and outcomes if they actually occurred?”)

and

https://www.ncbi.nlm.nih.gov/pmc/articles/PMC3853868 (here, please refer to point 7 “Failure to explicitly develop inclusion and exclusion criteria”, where the authors state “Suggestions for exclusion criteria include sufficient lack of variables recorded in the chart, …”).

Sincerely,

The corresponding author.

Reviewer 2 Report

In the manuscript “Use of antibiotics in poisonous ingestions of corrosives and or- 2

ganophosphates: A retrospective cohort study” (toxics-2239353), the authors explored the relationship between clinical outcomes and using antibiotics in acute poisonous inges- 16 tion involving corrosives or organophosphates. However, the study has significant criticisms (see below) that need to be addressed before publication. 

1. The relevant mechanistic relationships should be shown in the form of pictures, conjectures or inferences

2. The article has a single data structure and cannot be considered a research paper, it should be extended to cases or other literature for a comprehensive analysis

Author Response

Thank you for reviewing our manuscript and for your valuable comments, which are addressed below.

  1. The relevant mechanistic relationships should be shown in the form of pictures, conjectures or inferences.

Response: To address this comment, we drew manually the mechanism of organophosphate poisoning which happens through the inhibition of acetylcholine esterase. To better illustrate this mechanism, we drew first the signaling of acetylcholine, then the termination of its action by acetylcholine esterase, and then finally the inhibition of the latter by organophosphates. The illustration was added as figure 1 on page 2 and was cited in the introduction on line 54.

  1. The article has a single data structure and cannot be considered a research paper, it should be extended to cases or other literature for a comprehensive analysis.

Response: As this was a single-center study, we acknowledged the small sample size in the limitations section. However, we would like to note that we did in fact run sample size calculation and found that a samples size of only 94 patients was sufficient to meet a study power of 80% (which is the minimum in medical studies (https://www.ncbi.nlm.nih.gov/pmc/articles/PMC3409926/ “The ideal power for any study is considered to be 80%”) with an effect size of 30%. The details of the sample size calculation are listed under “Statistical analysis” in the methods section with a justification for the effect size (lines 281-286).

Regarding the referral to other literature, we did a literature search to identify any previously published similar studies and we could only identify one study by Priyendu, et al (reference #2); hence, we used to compare our findings with their findings in the discussion section to allow for a more comprehensive evaluation and understanding of our findings (lines 153-173).

Sincerely,

The corresponding author